# Does Androgen Deprivation for Prostate Cancer Affect Normal Adaptation to Resistance Exercise?

**DOI:** 10.3390/ijerph19073820

**Published:** 2022-03-23

**Authors:** Tormod S. Nilsen, Sara Hassing Johansen, Lene Thorsen, Ciaran M. Fairman, Torbjørn Wisløff, Truls Raastad

**Affiliations:** 1Institute of Physical Performance, The Norwegian School of Sport Sciences, PB 4014, 0807 Oslo, Norway; sarahj@nih.no (S.H.J.); trulsr@nih.no (T.R.); 2National Resource Centre for Late Effects, Department of Oncology, Oslo University Hospital, PB 4950, 4950 Oslo, Norway; lka@ous-hf.no; 3Department of Clinical Service, Division of Cancer Medicine, Oslo University Hospital, PB 4950, 4950 Oslo, Norway; 4Department of Exercise, University of South Carolina, Columbia, SC 29208-0001, USA; cfairman@mailbox.sc.edu; 5Health Services Research Unit, Akershus University Hospital HF, PB 1000, 1478 Lørenskog, Norway; twisloff@gmail.com

**Keywords:** adaptation, resistance exercise, prostate cancer patients, androgen deprivation therapy, healthy elderly men

## Abstract

Background: Loss of muscle mass and muscle function is a common side effect from androgen deprivation therapy (ADT) for prostate cancer (PCa). Here, we explored effects of heavy-load resistance training (RT) on lean body mass and muscle strength changes reported in randomized controlled trials (RCTs) among PCa patients on ADT and in healthy elderly men (HEM), by comparison of results in separate meta-analysis. Methods: RCTs were identified through databases and reference lists. Results: Seven RCTs in PCa patients (*n* = 449), and nine in HEM (*n* = 305) were included. The effects of RT in lean body mass change were similar among PCa patients (Standardized mean difference (SMD): 0.4, 95% CI: 0.2, 0.7) and HEM (SMD: 0.5, 95% CI: 0.2, 0.7). It is noteworthy that the within group changes showed different patterns in PCa patients (intervention: 0.2 kg; control: −0.6 kg) and HEM (intervention: 1.2 kg; control: 0.2 kg). The effects of RT on change in muscle strength (measured as 1 RM) were similar between PCa patients and HEM, both for lower body- (PCa: SMD: 1.9, 95% CI: 1.2, 2.5; HEM: SMD: 2.2, 95% CI: 1.0, 3.4), and for upper body exercises (PCa: SMD: 2.0, 95% CI: 1.3, 2.7; HEM: SMD: 1.9, 95% CI: 1.3, 2.6). Conclusions: The effects of RT on lean body mass and 1 RM were similar in PCa patients on ADT and HEM, but the mechanism for the intervention effect might differ between groups. It seems that RT counteracts loss of lean body mass during ADT in PCa patients, as opposed to increasing lean body mass in HEM.

## 1. Introduction

Androgen deprivation therapy (ADT) leads to castrate levels of testosterone and reduces cancer-related mortality in prostate cancer (PCa) patients [1,2,3]. Despite its effectiveness in treating PCa, ADT is often associated with side effects, such as loss of lean body mass, reduced muscle strength, and increased fat mass [4]. Given the higher age of many men with PCa, associations between low muscle mass and poor survival have also been reported [5,6]. More commonly, PCa patients on ADT are at increased risk of sarcopenic obesity [7,8], diabetes and cardiovascular disease [9,10,11]. Consequently, this emphasizes the need for strategies to offset the detrimental effects of ADT, particularly on body composition.

Heavy-load resistance training (RT), typically described as using external loads greater than 60% of their maximal strength lifted to near voluntary muscular failure [12], has been suggested as an adequate intervention to counteract the detrimental effects of ADT on lean body mass and muscle strength [13]. Several randomized controlled trials (RCTs) have demonstrated the effects of RT on muscle strength in PCa patients on ADT [14,15]. However, the effects of RT on increasing lean body mass during ADT is more uncertain [16,17]. This is also in agreement with RT interventions in other cancer types, indicating that individuals with cancer in general experience a blunted response in lean body mass in response to RT interventions [18]. However, whether PCs patients on ADT can expect similar effects of RT as healthy elderly men (HEM) is currently not known.

Therefore, the aim of the present meta-analysis was to explore effects of RT on lean body mass and muscle strength in PCa patients during ADT and in HEM, and to discuss potential differences.

## 2. Materials and Methods

We conducted a systematic review and meta-analysis to evaluate the impact of RT on lean body mass in PCa patients and in apparently healthy older adults.

### 2.1. Eligibility Criteria

Eligibility criteria for studies in PCa patients were men with PCa currently treated by ADT. Eligibility criteria for HEM were a cohort of men with a mean age of ≥50 years. Additional eligibility criteria for both populations were: (1) a RCT; (2) reported effects of RT on lean body mass and muscle strength; (3) RT included full-body training programs, using exercise loads ≥60% of one-repetition maximum (1 RM) (i.e., the maximal load that can be lifted once in a specific exercise), performed under supervision in gym-settings; and (4) included a non-exercise control group/wait-list control group. Furthermore, studies were excluded if the study participants received any supplements (e.g., protein supplement, creatine monohydrate, growth hormones, etc.) or were written in a non-English language.

Studies in HEM were excluded if: (1) results for the male participants were not reported separately (when studies included both sexes), or if authors were unable to provide separate results from men upon request; (2) participants had any specified disorders or conditions (3) participants received hormone replacement therapy; (4) participants performed >1 weekly RT sessions prior to the study (as this is a common exclusion criterion in studies among PCa patients).

### 2.2. Information Sources

Searches were performed in PubMed and AMED, from their inception date to December 2021

### 2.3. Search

We used the following search terms to identify studies in PCa patients: “Prostate cancer” AND (“exercise” OR “training”) AND (“weight lifting” OR “resistance training” OR “strength training”) AND (“muscle mass” OR “lean body mass” OR “lean mass”).

To identify eligible studies in HEM, we used the following search terms: elderly AND (exercise OR training) AND (“weight lifting” OR “resistance training” OR “strength training”) AND (“muscle mass” OR “lean body mass” OR “lean mass”).

In addition, manual searches of reference lists of original studies and relevant review articles were performed to identify additional studies. Abstracts from scientific conferences were not examined because of the paucity of requisite data.

### 2.4. Study Selection

After the search, two assessors (TSN and SHJ) independently removed duplicates, and screened all titles, abstracts, and full texts to identify eligible studies. A third researcher (TR) finally determined any divergences. The selection process was documented by Preferred Reporting Items for Systematic Reviews and Meta-analyses (PRISMA) [19].

### 2.5. Data Collection Process and Data Items

TSN and SHJ extracted data from all included studies independently. The extracted data from the two authors where then reviewed and merged. Any disagreement between assessors was corrected. The following data were extracted from all included studies: general characteristics (e.g., first author, publication year), participant information (e.g., sample size, age), intervention information (e.g., intervention duration, resistance training characteristics, and any other exercise content), and lastly the specific outcomes of intertest for this meta-analysis: lean body mass was assessed by dual x-ray absorptiometry (DXA) or hydrodensitometry. Muscle strength was assessed by 1 RM in lower- and/or upper body exercises. Lean body mass and 1 RM means and standard deviations were extracted for the exercise groups and control groups at baseline and post-intervention from all the included studies.

### 2.6. Risk of Bias in Individual Studies

The risk of bias in the included studies was assessed by the Cochrane Collaboration Tool for assessing the risk of bias in RCTs [20]. The risk assessment tool identifies seven areas that constitute potential sources for risk of bias in clinical trials: generation of allocation sequence, concealment of the allocation sequence, blinding of both study participants and study assessors, outcome data, selective reporting of data, and other threats to validity. Blinding of participants was not included in the present assessment, as participants are randomized to RT. The included studies were evaluated by “yes” (+), “no” (−), or “unclear” (?).

### 2.7. Summary Measures and Synthesis of Results

Studies were combined using random-effects meta-analysis with inverse variance weighting. For each study, the change from baseline to post-intervention was compared between the intervention and the control. Analyses are presented as forest plots of standardized mean difference (SMD) using Hedges’ g [21]. All analyses were performed using the package “meta” in R version 4.0.0 through RStudio Version 1.2.5042. Finally, comparisons of weighted mean changes of the intervention and control groups of PCa trials were compared to intervention and control groups in HEM trials using t-tests for two independent samples.

## 3. Results

After the removal of duplicates, 137 publications in PCa patients were screened, of which 131 were excluded (Figure 1a). Reasons for exclusions were interventions other than RT (e.g., testosterone supplementation, dietary interventions, cancer prevention, etc.) (*n* = 50), not a randomized controlled design (*n* = 30) or other cohorts than PCa patients (*n* = 23). Additional reasons are listed in Figure 1a. The remaining six studies, and one study identified from reference lists, were included in the meta-analysis.

The search strategy yielded 698 publications in HEM, of which 690 were excluded (Figure 1b). Major reasons for exclusions were not a randomized controlled design (*n* = 213), studies in other cohorts than HEM (*n* = 128), and studies in women (*n* = 125). Additional reasons are listed in Figure 1b. For studies including both men and women, the corresponding authors were contacted to request separate results for male participants. In eight cases of this nature, we were able to obtain separate results for males for six studies, and these studies were included. The remaining two studies were excluded, as we were not able to obtain separate results for male participants. Three additional studies were included after manual inspection of reference lists.

### 3.1. Risk of Bias

All studies had high risk of bias for blinding of participants, as it is impossible to blind study participants to exercise or no exercise (Figure 2). Overall, a lower risk of bias was observed in studies among PCa patients than in studies among HEM. Specifically, HEM had a higher proportion of studies with unclear or random sequence generation and allocation concealment. Detailed descriptions of risk of bias assessments are presented as Appendix A.

### 3.2. Characteristics of Included Studies

PCa patients: The seven included studies were published between 2010 and 2019 and included a total of 449 patients, 230 and 219 in the intervention and control groups, respectively (Table 1). The number of patients in the intervention groups ranged from 23 to 57 and from 22 to 50 in the control groups. Intervention duration ranged from 12 to 52 weeks. Four of the seven interventions consisted of three weekly sessions and two weekly sessions in the remaining three. The total number of sessions ranged from 24 to 156. The training loads ranged from six to 12 repetitions maximum (RM), in one to three sets. Two of the six interventions also included an aerobic component, consisting of 20 to 30 min of moderate, continuous endurance training.

HEM: The nine studies were published between 1996 and 2019 and included a total of 291 men, 154 and 151 in the intervention- and control groups, respectively (Table 2). The number of participants in the intervention groups ranged from 6 to 46, and from 5 to 44 in the control groups. Intervention duration ranged from 12 to 48 weeks. Seven of the nine interventions consisted of three weekly sessions, and two weekly sessions in the remaining two. The total number of sessions ranged from 24 to 156. The training loads ranged from 60% to 90% of 1 RM, and from 6–12 RM, in one to three sets. One of the interventions included aerobic training.

### 3.3. Effect of Resistance Training on Lean Body Mass

PCa patients: In PCa patients, lean body mass increased by a weighted average of 0.2 kg in the exercise groups, following RT, whereas a −0.6 kg reduction was reported in the control groups (Table 3). There was a significant difference in lean body mass change over the intervention period between exercise- and control groups (SMD: 0.4 (95% CI: 0.2, 0.7)) (Figure 3a).

HEM: In HEM, lean body mass increased by a weighted average of 1.2 kg in the exercise groups following RT, whereas lean body mass remained unchanged in the control groups (0.1 kg) (Table 3). There was a significant difference in lean body mass change over the intervention period between exercise- and control groups (SMD: 0.5, (95% CI: 0.2, 0.7)) (Figure 3b).

### 3.4. Effect of Resistance Training on Muscle Strength

#### 3.4.1. Lower Body Exercises—PCa Patients

Lower body 1 RM increased by a weighted average of 20.6 kg in the exercise groups following RT, whereas 1 RM remained unchanged in the control groups (1.8 kg) (Table 3). There was a significant difference in lower body 1 RM change over the intervention period between exercise- and control groups (SMD: 1.9 (95% CI: 1.2, 2.5)) (Figure 4a).

#### 3.4.2. Lower Body Exercises—HEM

Lower body 1 RM increased by a weighted average of 37.4 kg in the exercise groups following RT in HEM, whereas 1 RM remained almost unchanged in the control groups (1.8 kg) (Table 3). There was a significant difference in lower body 1 RM change over the intervention period between exercise- and control groups (SMD: 2.2, (95% CI: 1.0, 3.4)) (Figure 4b).

#### 3.4.3. Upper Body Exercises—PC Patients

Upper body 1 RM increased by a weighted average of 6.0 kg in the exercise groups following RT, whereas 1 RM remained unchanged in the control groups (−0.1 kg) (Table 3). There was a significant difference in upper body 1 RM change over the intervention period between the exercise and control groups (SMD: 2.0 (95% CI: 1.3, 2.7)) (Figure 4c).

#### 3.4.4. Upper Body Exercises—HEM

Upper body 1 RM increased by a weighted average of 13.2 kg in the exercise groups in HEM, whereas 1 RM remained unchanged in the control groups (−2.3 kg) (Table 3). There was a significant difference in lean upper body 1 RM change over the intervention period between the exercise and control groups (SMD: 1.9, (95% CI: 1.3, 2.6)) (Figure 4d).

#### 3.4.5. Difference in Within-Group Changes between PCa and HEM

The weighted mean lean body mass changes across all studies revealed declines in the control groups of PCa patients (−0.6 kg, (95% CI: −1.0, −0.2)), while there was a small increase in HEM in the control groups (0.2 kg, (95% CI: −0.6, 1.0)) (Table 3). A post-hoc independent samples t-test comparing the changes between the control groups (i.e., PCa patients vs. HEM) showed significant differences (*p* = 0.048) (Table 3). While the average lean body mass slightly increased following RT in PCa (0.2 kg, (95% CI: −0.2, 0.6)), there was a larger increase in the intervention groups of HEM (1.2 kg, (95% CI: 0.5, 1.8) (Table 3). When comparing changes between PCa patients and HEM, a significant difference in mean change was observed (*p* = 0.008) (Table 3).

Similar changes in 1 RM were observed within the control- and intervention groups for both upper- and lower body exercises, with no significant differences between PCa patients and HEM (Table 3).

## 4. Discussion

The findings of this meta-analysis demonstrate that participation in RT results in significant effects on lean body mass and muscle strength in both PCa patients on ADT and in HEM, but the mechanisms behind the intervention effect might differ between PCa patients and HEM. It seems that RT counteracts the loss of lean body mass during ADT in PCa patients, whereas participation in RT results in an increase in lean body mass following the intervention in HEM. Thus, while HEM can expect an increase in lean body mass with an intervention, a positive outcome for PCa could very well be the preservation of LBM, where a decrease in lean body mass would otherwise be experienced. Nevertheless, the intervention effects reported here are identical between PCa patients and HEM, indicating that both groups benefit from participating in RT. This is also supported by the similar increases in muscle strength experienced from participation in RT in both PCa patients and HEM.

Such findings are of clinical relevance, especially in PCa patients in ADT, where declines in lean body mass and muscle strength are associated with impaired glycemic control [34], increased risk of cardiovascular comorbidities, as well as increased risk of falls and fractures [35,36], which are established late effects from ADT [37,38,39]. Importantly, impairments of physical performance during ADT may not recover even years after treatment cessation [40]. Consequently, RT in concurrence with ADT may provide a preventive strategy to maintain muscle mass, strength and overall health-related quality of life. Importantly, this knowledge could also be important to practitioners delivering an RT program to PCa patients. Findings from the behavioral literature indicate that individual values, perceived benefits and expected outcomes, can influence the adoption of health behaviors [41,42]. Consequently, knowledge of expected outcomes from RT could help practitioners appropriately guide PCa patients’ expectations regarding anticipated outcomes.

The reasons for a blunted response to RT during ADT are currently unknown, as there is limited information on the mechanisms behind ADT-induced lean body mass loss [43]. To the best of our knowledge, the effect of RT on lean body mass has only been compared directly in PCa patients on and off ADT in one study, and no difference between the groups was observed (per reported confidence intervals for the changes) [22]. However, our findings are in line with observations in young, healthy men randomized to receive ADT (Goserelin) or placebo (saline) during 12 weeks of RT, where significantly greater increases in lean body mass was observed in the placebo group [44].

Hanson et al. (2017) reported lower baseline muscle protein synthesis rates in PCa patients on ADT compared to age-matched, non-hypogonadal men [45]. However, similar increases in the rate of muscle protein synthesis following an acute bout of RT between the PCa patients on ADT and the reference group were reported [45]. How participation in RT on a more regular basis affects muscle protein synthesis rates is currently not known, but an increased cross-sectional area of type II muscle fibers has been reported [46], indicating the accumulation of muscle proteins. Furthermore, the RT prescriptions evaluated by the studies included in the present meta-analysis are relatively homogenous in terms of intervention duration, weekly training frequency and training load. A non-controlled study by Hanson et al. (2013) showed greater lean body mass increases compared to the RCTs included in the present meta-analysis [47]. Interestingly, Hanson et al. [47] used drop-sets in their RT prescription, which normally results in a larger session training volume (i.e., training load (kg) x number of repetitions x number of sets) compared to more conventional RT used in the studies included in the present meta-analysis. Thus, research into RT variables, such as training frequency, load, and volume, is needed to optimize RT guidelines during ADT. Furthermore, knowledge of nutritional strategies to optimize RT-induced effects on lean body mass is currently scarce, but clinical trials are currently underway [48,49,50]. Specifically, creatine supplementation might lead to an increased training volume [51], and optimizing participants diets by ensuring sufficient protein content [52] holds the potential to further improve lean body mass responses to RT.

Our findings among PCa patients on ADT are in accordance with a previous meta-analysis. Keilani et al. (2017) performed a meta-analysis amongst all prospective RT trials in PCa patients on ADT on several exercise-induced outcomes [13]. The authors identified seven papers reporting on the effects of RT on lean body mass and that participation in RT resulted in significant improvements in lean body mass (~1 kg; CI [0.15, 1.84]) [13]. However, our meta-analysis expands on the work by Keilani and co-workers [13] by including RCTs only, and thus taking the intervention effect (i.e., the difference in mean change between intervention groups and control groups) into account. Furthermore, the comparison of meta-analysis between the effects of RT observed in PCa patients and in HEM is another major advantage of the present study, as it provides perspective to the intervention effects seen amongst PCa patients on ADT, relative to those without cancer.

Interestingly, the application of ADT in PCa patients continues to evolve. Originally intended as lifelong therapy for advanced, metastatic cancer, ADT is now routinely used in non-metastatic disease [53]. Furthermore, the duration and continuity of ADT use is constantly developing [54]. Continuous ADT administration has typically been the convention, whereas intermittent ADT is now being routinely considered in patients with a lower burden of disease [54]. It follows, then, that the specific application of ADT (i.e., duration, intermittent vs. continuity), may influence the degree of testosterone suppression and ultimately the response to RT. As these treatments continue to evolve, there is a need for concurrent research into the impact of new treatment regimens on body composition changes and the resultant ability to respond favorably to an RT stimulus.

Our meta-analysis suffers from limitations that need to be addressed. First, it is important to acknowledge that the studies included in the present meta-analysis, in general, consist of relatively small study samples and with interventions of relatively limited duration. Therefore, the external validity of our results may be limited, and any long-term effect of RT on lean body mass might have been missed. Second, we might have missed some studies despite our efforts and search strategies. Searches in different databases and duplicate screenings were applied to reduce the number of missed studies. Finally, we will highlight the fact that the meta-analysis on HEM may not be representative for the average population, given the restrictions that led to the exclusion of most studies that were identified by our searches. However, the aim of the present meta-analysis was not to perform a comprehensive overview of RT effects in HEM, and the restrictions were applied to make the comparison with PCa patients as valid as possible.

## 5. Conclusions

Heavy-load resistance training leads to significant between-group changes in lean body mass in both PCa patients on ADT and in HEM. However, counteracting the loss of muscle mass observed within the control groups seems to be driving the intervention effect amongst PCa patients, whereas increases in lean body mass in the intervention groups seems to be driving the effect in HEM. Importantly, our observations should be validated by trials directly comparing effects of RT between PCa patients and HEM.

## Figures and Tables

**Figure 1 ijerph-19-03820-f001:**
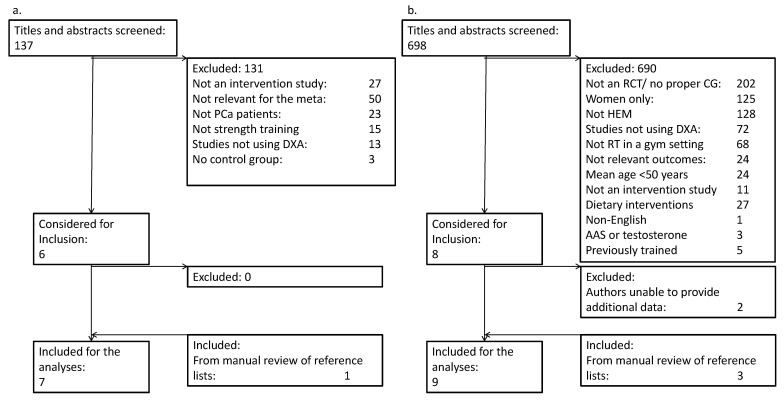
PRISMA flow diagram for (**a.**) prostate cancer patients and (**b.**) healthy elderly men.

**Figure 2 ijerph-19-03820-f002:**
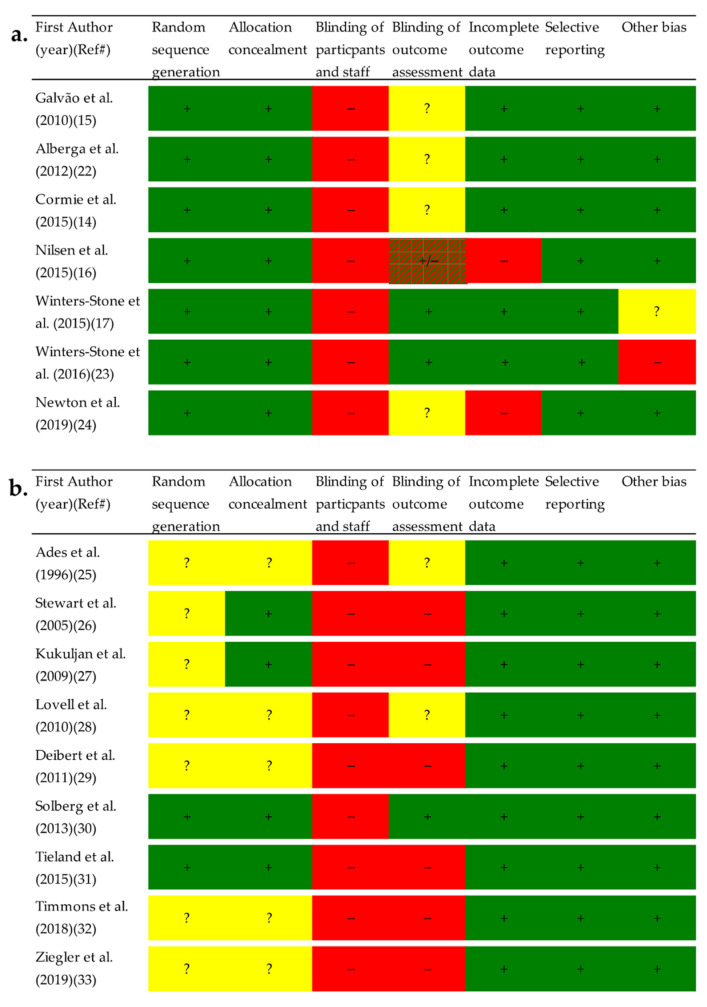
Risk of bias assessment for (**a.**) Prostate cancer patients and (**b.**) Healthy elderly men. Each risk of bias item in the included studies were evaluated as meeting the criteria for low risk of bias with “yes” (+), “no” (−), or “unclear” (?). Abbreviation: Ref#, Reference number.

**Figure 3 ijerph-19-03820-f003:**
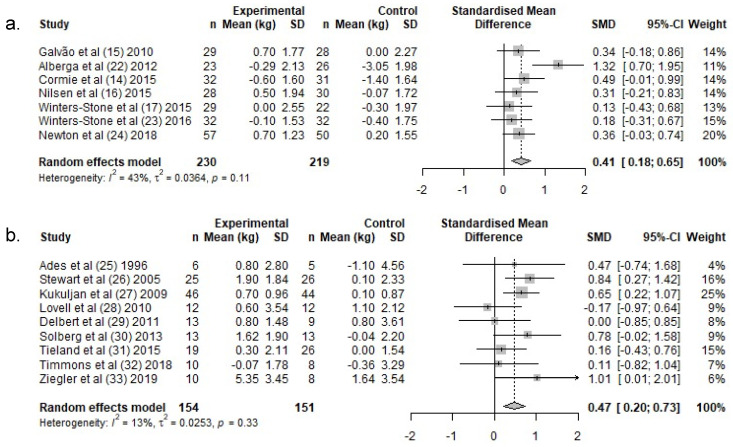
Meta-analysis for change in lean body mass for (**a.**) PCa and (**b.**) HEM. Abbreviations: *n*, Number; SD, Standard deviation; SMD, standardized mean difference; CI, confidence intervals.

**Figure 4 ijerph-19-03820-f004:**
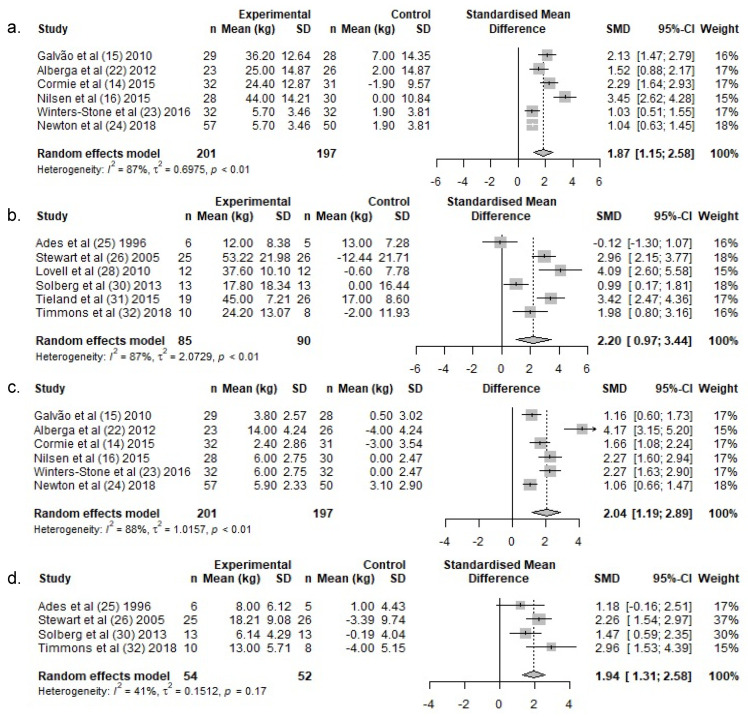
Meta-analysis for change in muscle strength for lower-body exercises for (**a.**) PCa and (**b.**) HEM, and in upper-body exercises for (**c.**) PCa and (**d.**) HEM. Abbreviations: *n*, Number; SD, standard deviation; SMD, standardized mean difference; CI, confidence intervals.

**Table 1 ijerph-19-03820-t001:** Study characteristics in prostate cancer patients.

First Author	Study Arms	*n*	Weeks	Sessions/Week	Sessions in Total	Strength Training Program	Aerobic Training Program
Galvão et al. (2010) [15]	EG CG	29 28	12	2	24	Full body program, 1–4 sets of 6–12 RM	20–30 min at 70–85% of HR_max_
Alberga et al. (2012) [22]	EG CG	23 26	24	3	72	Full body program, 1–2 sets of 8–12 rep at 60–70% of estimated 1 RM	N.A.
Cormie et al. (2015) [14]	EG CG	32 31	12	2	24	Full body program, 1–4 sets of 6–12 RM	20–30 min at 70–85% of HR_max_ Encouraged to engage in 150 min of home-based, aerobic exercise
Nilsen et al. (2015) [16]	EG CG	28 30	16	3	48	Full body program, 1–3 sets of 6–10 RM	N.A.
Winters-Stone et al. (2015) [17]	EG FLEX *	29 22	52	3 (of which 1 was home-based)	156	Full body program, 1–2 sets of 8–12 reps at 60–80% of 1 RM	N.A.
Winters-Stone et al. (2016) [23]	EG CG	32 32	24	3	72	Full body program, 1–2 sets of 8–12 reps at 60–80% of 1 RM **	N.A.
Newton et al. (2019) [24]	EG CG	57 50	24	2	48	A combined intervention of impact exercises and resistance exercise. Full body program ***, 2–4 sets of 6–12 RM	N.A.

* Winters-Stone et al. (2015) included stretching as an attention-control group. ** With the addition of exercises that couples could do together. *** Exercise selection was customized to avoid parts of the body affected by metastasis. Abbreviations: *n*, number; EG, Exercise group; CG, Control group; Reps, Repetitions; RM, Repetition maximum (i.e., the maximum weight that can lifted in a given number of repetitions); HR_max_, Maximal heart rate; N.A., Not applicable (not included in the training program); 1 RM, one repetition maximum (i.e., maximal strength: the maximum weight that can be lifted once in a given exercise).

**Table 2 ijerph-19-03820-t002:** Study characteristics in healthy elderly men.

**First Author**	**Study Arms**	** *n* **	**Weeks**	**Sessions/Week**	**Sessions in Total**	**Strength Training Program**	**Aerobic Training Program**
Ades et al. (1996) [25]	EG CG	6 5	12	3	36	Full body program, three sets of eight reps at 80% of 1 RM	N.A.
Stewart et al. (2005) [26]	EG CG	25 26	26	3	78	Full body program, two sets of 10–15 reps with progressing load	45 min on an endurance ergometer at 60–90% of HR_max_
Kukuljan et al. (2009) [27]	EG CG	46 44	48	3	144	Full body program, 8–12 reps at 60–85% of 1 RM	N.A.
Lovell et al. (2010) [28]	EG CG	12 12	16	3	48	Squat exercise only, three sets, 6–10 reps at 70–90% of 1 RM.	N.A.
Deibert et al. (2011) [29]	EG CG	13 9	12	2	24	Full body program 10 to 25 RM (sets not provided)	N.A.
Solberg et al. (2013) [30]	EG CG	13 13	12	3	36	Full body program, three sets of 6–12 RM	N.A.
Tieland et al. (2015) [31]	EG CG	19 26	24	2	48	Full body program, three sets of 8–15 reps at 50–75% of 1 RM	N.A.
Timmons et al. (2018) [32]	EG CG	10 8	12	3	36	Full body program, four sets of 15 RM *	N.A.
Ziegler et al. (2019) [33]	EG CG	10 8	52	3	156	Full body program 1–3 sets of 6–12 RM (after familiarization)	N.A.

* Workload started at 60% of 1 RM and was progressed by 5% when 15 repetitions could be performed with proper form. Abbreviations: *n*, number; EG, Exercise group; CG, Control group; Reps, Repetitions; 1 RM, one repetition maximum (i.e., maximal strength: the maximum weight that can lifted once in a given exercise); N.A., not applicable (not included in the training program); HR_max_, Maximal heart rate; RM, repetition maximum (i.e., the maximum weight that can lifted in a given number of repetitions).

**Table 3 ijerph-19-03820-t003:** Within-group changes and comparison of within-group changes between PCa patients and HEM.

	Prostate Cancer Patients					Healthy Elderly Men					PCa vs. HEM
	*n*	Weighted Mean	Pooled SD	CI Lower	CI Upper	*n*	Weighted Mean	Pooled SD	CI Lower	CI Upper	*p*-Value
Lean body mass (kg)											
Intervention	230	0.2	3.2	−0.2	0.6	154	1.2	4.0	0.5	1.8	0.008
Control	219	−0.6	3.3	−1.0	−0.2	151	0.2	4.8	−0.6	1.0	0.048
1 RM lower body exercises (kg)											
Intervention	201	20.6	107.5	5.8	35.5	85	37.4	247.8	−15.2	90.1	0.424
Control	197	1.8	96.4	−11.6	15.1	90	1.8	223.2	−45.7	49.2	0.999
1 RM upper body exercises (kg)											
Intervention	201	6.0	8.1	4.9	7.1	67	13.2	53.6	−1.1	27.5	0.067
Control	197	−0.1	9.7	−1.5	1.2	61	−2.3	59.0	−18.0	13.5	0.626

Abbreviations: *n*, Number; SD, Standard deviation; CI, Confidence intervals; PCa, Prostate cancer; HEM; Healthy elderly men; kg, Kilogram.

## Data Availability

Not applicable.

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
