# Peer review of "Does Androgen Deprivation for Prostate Cancer Affect Normal Adaptation to Resistance Exercise?"

_ijerph, 2022, doi:10.3390/ijerph19073820_

Round 1

Author Response

We want to thank reviewer 1 for a fair review that has helped improve our manuscript! Hopefully, we have been able to amend the manuscript accordingly. Below, we have pasted in the reviewer report, and our responses are listed in red. 

The paper is very interesting and important. The paper has the potential to publish at the above journal after the authors will address the following issues:

- At section 2: Material and method, it is missing a relevant reference.

  • We have cited three references under Material and methods. If the Reviewer suggests that we should cite some other references, we are happy to discuss this. However, we are not quite sure what the Reviewer is referring to here.

- After section 2, the authors present the results of the research. The authors must extend the method and be more specify to understand better the results section. They should describe in detail the methodology of the research.

  • We have added additional information on the data extraction and throughout the rest of the Methods section. We hope that this is sufficient to meet the Reviewers' comments.

- Nomenclature of parameters and abbreviation must be ad to the paper as well as units of dynamical variables.

  • Thank you for capturing this! We have added units to all tables, figures, and legends where relevant.

- The analysis of the figures must extend.

  • We have upgraded all table- and figure legends. We hope that this will fulfil the two suggestions above.

-The conclusion section as well as the discussion section must be extended for more details.

  • Thank you for the opportunity to extend our paper. We have updated the discussion and conclusion to some extent.

- The source of the data should cite in the table and in the paper.

  • We have inserted reference numbers to all tables and figures.

Reviewer 2 Report

Nilsen et al have written a fantastic paper that identifies the benefits of resistance training for PCa patients on ADT, albeit the prevention of worsening lean mass loss compared to increased lean mass in HEM. Please see my comments below. 

  1. Identify in the abstract that this was a systematic review/meta analysis. 
  2. Line 55: "HEM were men with a mean age of ≥50" - unclear if you mean to say a "cohort of HEM with a mean age ≥50 years," or "men aged ≥50 years." 
  3. Characterisation of studies: In the supplement file you identify that some studies included home-based components, however, this is not acknowledged when describing the study characteristics. Please add this into the text so that the reader may get a full description of what interventions were prescribed. 
  4. Characterisation of studies: in the HEM section you identify %RM, add this to the PCa paragraph if available. 
  5. Results: Lines 189, 198, 204, 210 = instead of "difference in lean body mass change" shouldn't it say "difference in upper/lower body muscle strength"?
  6. Results: you state "similar changes in 1RM were observed" while yes I agree similar patterns in terms of intervention versus control (which is what you state) were seen but add that the changes were not different between PCa and HEM, this was not similar to lean mass results. 
  7. I have some concern with the use of "heavy load" RT in the abstract, introduction, and conclusion given it is not defined nor is it explicitly mentioned throughout the rest of the paper. Please define what you mean by heavy load RT, and how you took that into account in your inclusion criteria.
  8. There has been some suggestion that including aerobic exercise, particularly in the same session as RT, may also blunt the intervention effect on muscle mass development. This may be worth acknowledging in the discussion given some studies included aerobic exercise. 
  9. You acknowledge nutritional strategies to optimize lean mass are scarce. I see the references noted here are creatine/protein based - I think it is worth elaborating on this section to explicitly acknowledge the potential role of these supplements, as opposed to stating just "nutritional strategies". 
  10. There are a number of writing errors in both the main text and supplementary file, please read through again to correct these. Lines of interest include: 45, 65, 219, 256. 

Author Response

We want to thank reviewer 2 for their excellent comments and suggestions. Below, we have pasted the reviewer report and you will find our responses in red underneath each suggestion. We hope that our responses are sufficient to meet the standard of reviewer 2. 

Nilsen et al have written a fantastic paper that identifies the benefits of resistance training for PCa patients on ADT, albeit the prevention of worsening lean mass loss compared to increased lean mass in HEM. Please see my comments below. 

  1. Identify in the abstract that this was a systematic review/meta analysis. 
  • Thank you for this suggestion. We added a sentence to the abstract.
  1. Line 55: "HEM were men with a mean age of ≥50" - unclear if you mean to say a "cohort of HEM with a mean age ≥50 years," or "men aged ≥50 years." 
  • Thank you for clearing this up. “Cohort of HEM with a mean age of ≥50 years” has been added to the manuscript in line 63.
  1. Characterisation of studies: In the supplement file you identify that some studies included home-based components, however, this is not acknowledged when describing the study characteristics. Please add this into the text so that the reader may get a full description of what interventions were prescribed. 
  • Thank you for this suggestion. We have added the home-based components of Cormie et al. (2015) and Winters-Stone et al. (2015) to the table.
  1. Characterisation of studies: in the HEM section you identify %RM, add this to the PCa paragraph if available. 
  • This is a great suggestion to make it easier to compare the exercise intervention across populations. However, given the considerable heterogeneity in the number of repetitions different individuals can perform at a given load relative to 1RM, we are not comfortable with extrapolating nRN to % of 1RM. Each study is presented as described by the authors, and we would be more comfortable with keeping it this way.

  1. Results: Lines 189, 198, 204, 210 = instead of "difference in lean body mass change" shouldn't it say "difference in upper/lower body muscle strength"?
  • Thank you for catching this! You are right! We have updated the manuscript accordingly!

  1. Results: you state "similar changes in 1RM were observed" while yes I agree similar patterns in terms of intervention versus control (which is what you state) were seen but add that the changes were not different between PCa and HEM, this was not similar to lean mass results. 
  • We have added that there were no differences between PCa and HEM to the sentence. However, please note that the study by Treuth has been removed from the upper body 1RM since the initially submitted version. The study was excluded earlier, but unfortunately, it was not removed from the last version.

  1. I have some concern with the use of "heavy load" RT in the abstract, introduction, and conclusion given it is not defined nor is it explicitly mentioned throughout the rest of the paper. Please define what you mean by heavy load RT, and how you took that into account in your inclusion criteria.
  • We appreciate this suggestion, as it helps define an important aspect of our paper. We have added a definition and a reference to the introduction and specified the exercise load under the inclusion criteria.

  1. There has been some suggestion that including aerobic exercise, particularly in the same session as RT, may also blunt the intervention effect on muscle mass development. This may be worth acknowledging in the discussion given some studies included aerobic exercise. 
  • This may be a valid discussion point since a larger proportion of PCa patients were exposed to aerobic exercise compared to the HEM (26.5% of PCa patients Vs 16.2% of HEM). However, the latest meta-analysis on the topic found that aerobic exercise did not blunt hypertrophic responses to resistance training when compared to resistance training alone, neither measured as muscle CSA nor as lean body mass (PMID 34757594). Furthermore, given the relatively small proportion of participants that were “exposed” to aerobic exercise, we find it unlikely that aerobic exercise could explain the difference in hypertrophic responses to resistance training as reported in our paper.

  1. You acknowledge nutritional strategies to optimize lean mass are scarce. I see the references noted here are creatine/protein based - I think it is worth elaborating on this section to explicitly acknowledge the potential role of these supplements, as opposed to stating just "nutritional strategies". 
  • Thank you for this suggestion. We have elaborated on creatine- and protein supplementation as strategies to potentially enhance LBM increases from resistance training.
  1. There are a number of writing errors in both the main text and supplementary file, please read through again to correct these. Lines of interest include: 45, 65, 219, 256. 
  • Thank you for highlighting this. We hope we have corrected most errors in this revision.

Reviewer 3 Report

Thank you for the possibility of reviving the manuscript entitled  “Does androgen deprivation for prostate cancer affect normal adaptation to resistance exercise?”. The authors focused on a very important topic, and I appreciate their work on this manuscript. Overall, data is presented understandably. I have only a few comments on the abstract structure and improvements of the figure legend so that readers would benefit more from this work.

- please include in the Abstract at least a sentence of the general introduction of the topic. Moreover, please avoid too many numbers in the abstract and instead focus on words.

- please add a legend in Figure 2 for a denotes like ? , -, +

- Please improve the first column in Figure 2, Table 2, Figure 3, adding reference number

- please explain all aberrations used in the Figures/Tables in their legends/ footers

Author Response

We want to thank reviewer 3 for their great effort to improve our manuscript! Please find our responses to Reviewers' comments in red under each suggestion made by the reviewer. 

Thank you for the possibility of reviving the manuscript entitled  “Does androgen deprivation for prostate cancer affect normal adaptation to resistance exercise?”. The authors focused on a very important topic, and I appreciate their work on this manuscript. Overall, data is presented understandably. I have only a few comments on the abstract structure and improvements of the figure legend so that readers would benefit more from this work.

- please include in the Abstract at least a sentence of the general introduction of the topic. Moreover, please avoid too many numbers in the abstract and instead focus on words.

  • Thank you for this suggestion. We added a sentence to provide the background for our aims. We tried to revise the Abstract without numbers but found it hard to disseminate our findings appropriately. Therefore, we would like to keep the numbers in. 

- please add a legend in Figure 2 for a denotes like ? , -, +

  • Thank you for this suggestion. We have added an explanation to the legend.

- Please improve the first column in Figure 2, Table 2, Figure 3, adding reference number

  • We have inserted reference numbers to all figures and tables.

- please explain all aberrations used in the Figures/Tables in their legends/ footers

  • Thank you for bringing this to our attention. We hope we have captured all abbreviations used in the figures and tables.

Round 2

Reviewer 1 Report

The authors did not add the reference as I request in the first round of revision:

Method of directly defining the inverse mapping applied to prostate cancer immunotherapy Mathematical model. O Nave, M Elbaz. International Journal of Biomathematics

Author Response

The authors did not add the reference as I request in the first round of revision:

Method of directly defining the inverse mapping applied to prostate cancer immunotherapy Mathematical model. O Nave, M Elbaz. International Journal of Biomathematics

  • Dear Reviewer 1: Thank you for this suggestion. However, we couldn’t see how this reference would fit into the present manuscript. The suggested paper was circulated amongst the author group, and none of us could see the relevance to this meta-analysis. Primarily since the article by Nave & Elbaz deals with other treatment modalities than included in this paper (i.e., immunotherapy, as opposed to androgen deprivation therapy in the present meta-analysis). However, we welcome your thoughts on where and how the reference should be included if you still think that the reference should be included.